# Phase-Matching Gating for Isolated Attosecond Pulse Generation

Vasily V. Strelkov [1,2,*] and Margarita A. Khokhlova [3]

1  Prokhorov General Physics Institute of the Russian Academy of Sciences, Vavilova Street 38, Moscow 119991, Russia
2  Moscow Institute of Physics and Technology, National Research University, Dolgoprudny 141700, Russia
3  Attosecond Quantum Physics Laboratory, Department of Physics, King's College London, Strand Campus, London WC2R 2LS, UK; margarita.khokhlova@kcl.ac.uk
*  Correspondence: strelkov.v@gmail.com

**Abstract:** We investigate the production of an isolated attosecond pulse (IAP) via the phase-matching gating of high-harmonic generation with intense laser pulses. Our study is based on the integration of the propagation equation for the fundamental and generated fields with nonlinear polarisation found via the numerical solution of the time-dependent Schrödinger equation. We study the XUV energy as a function of the propagation distance (or the medium density) and find that the onset of the IAP production corresponds to the change from linear to quadratic dependence of this energy on the propagation distance (or density). Finally, we show that the upper limit of the fundamental pulse duration for which IAP generation is feasible is defined by the temporal spreading of the fundamental pulse during the propagation. This nonlinear spreading is defined by the difference in the group velocities for the neutral and photoionised medium.

**Keywords:** attosecond physics; high-harmonic generation; intense laser field; attosecond pulses; isolated attosecond pulse; phase matching; gating techniques; propagation; time-dependent Schrödinger equation





## 1. Introduction

One of the central directions of attosecond physics [1–4] remains the advance of the attosecond pulses generated via the process of high-harmonic generation (HHG), typically in the shape of an attosecond pulse train. Attosecond XUV pulses have already proven to be a unique tool which are used to study ultrafast electronic processes in atoms, molecules and solids at their natural time scale. Among improvements in the attosecond pulses' duration, making them shorter to increase the temporal resolution of the processes, and in their intensity, making them brighter to cover a wider range of processes, there is an important goal: the creation of isolated attosecond pulses (IAPs) [5]. IAPs combined with intense laser pulses provide a very powerful pump–probe technique, which can be exploited for attosecond streaking [6,7] and transient-absorption spectroscopy [8–10]. Recently, the high demand for the generation of IAPs was emphasised by the realisation of attosecond-pump attosecond-probe experiments [11], which rely on high XUV energies to work.

The generation of IAPs via HHG has been demonstrated using a number of gating schemes [12]: amplitude gating (where typically a few-cycle laser pulse provides the gating) [13–15], polarisation gating (first demonstrated in [16] and later in [17,18]), double optical gating [19,20], spatio-temporal gating (the attosecond lighthouse technique [21–23] and noncollinear gating [24]), and phase-matching gating [25–29] (also called ionisation gating in cases of the microscopic effect). In this paper, we focus on phase-matching gating.

The phase-matching gating scheme is based on the creation of a temporal window for macroscopically propagating HHG emissions due to the compensation of the phase mismatch of contributions coming from the free electrons and from the neutral atoms [30,31]. As long as the relative contribution changes with the ionisation degree during the laser

pulse, then this finite temporal window can be achieved. Moreover, this window can be short enough to fit only a single attosecond pulse, or an IAP.

In this paper, we perform a numerical study of macroscopic HHG in argon gas by solving the 3D time-dependent Schrödinger Equation (TDSE) coupled with the reduced propagation Equation [30,31]. We choose to use the intense laser pulse in a way that results in the generation of an IAP within phase-matching gating. We calculate the propagation of IAPs generated in gas for different laser pulse parameters—the pulse intensity and the pulse duration—and we find the optimal regime for the IAP generation.

## 2. Methods

The following length scales characterising macroscopic aspects of HHG were introduced already in the early studies of the process [32]:

(i)　The coherence length

$$L_{\mathrm{coh}} = \pi / |\Delta k|,\tag{1}$$

where $\Delta k = k_q - q k_0$ is the phase mismatch for the generation of the $q$th harmonic; $k_q$ and $k_0$ are the harmonic and fundamental wavevectors, respectively; and

(ii)　The absorption length

$$L_{\mathrm{abs}} = 1 / (\sigma N^{\mathrm{at}}),\tag{2}$$

where $N^{\mathrm{at}}$ is the gas density and $\sigma$ is the ionisation cross section for the harmonic field.

However, it is important to take into account that the strong laser field causes ionisation of the medium during the generation, which results in the coherence length significantly varying in time within the laser pulse duration. In this case, the macroscopic HHG properties are characterised by another length scale, namely the blue-shift length [30]

$$L_{\mathrm{bs}} = \frac{\pi c}{q \omega_0 |n_{\mathrm{f}} - n_{\mathrm{i}}|},\tag{3}$$

where $n_{\mathrm{i}}$ and $n_{\mathrm{f}}$ are the fundamental refractive index at the beginning and end of the generation, respectively. As the variation in the refractive index is caused by the gas ionisation, to calculate $L_{\mathrm{bs}}$ we find the ionisation probability $w$ after the pulse (it almost does not change with the propagation distance for the distances considered here). Then, we obtain

$$|n_{\mathrm{f}} - n_{\mathrm{i}}| = 4.5 \times 10^{-22} \, w \, N^{\mathrm{at}} [\mathrm{cm}^{-3}] \lambda_0^2 [\mu\mathrm{m}],\tag{4}$$

where $N^{\mathrm{at}}$ is the initial atomic density (typically of the order of $10^{18} \, \mathrm{cm}^{-3}$) and $\lambda_0$ is the driver wavelength.

To simulate the macroscopic harmonic response, we numerically integrate the 1D propagation equation, and at each step of the propagation we calculate the nonlinear polarisation of the medium by solving the 3D TDSE for a model argon atom (for more details, see [30,31]). For HHG via a spatial flat-top beam [27,33,34], the laser intensity is almost constant up to a certain distance from the beam axis, so the 1D propagation equation is adequate.

In more detail, we start from the wave equation

$$\left( \frac{\partial^2}{\partial x^2} - \frac{1}{c^2} \frac{\partial^2}{\partial t^2} \right) E(x,t) = \frac{4\pi}{c^2} \frac{\partial^2}{\partial t^2} P(x,t)\tag{5}$$

for the field $E(x,t)$ linearly polarised along the $z$-axis and propagating along the $x$-axis through a medium with nonlinear polarisation $P(x,t)$.

Wave Equation (5) can be simplified [35] to the reduced first-order wave equation by applying the slowly varying amplitude approximation under the conditions of modest

medium density and laser intensity, which are discussed in detail in [36,37]. If we define the Fourier transform for some function $G(x,t)$ as

$$G(x,\omega) = \int\limits_{-\infty}^{\infty} G(x,t)\exp i\omega t dt \tag{6}$$

and slowly varying (along propagation coordinate $x$) amplitudes $\tilde{G}(x,\omega)$ of the spectral components $G(x,\omega)$ as

$$G(x,\omega) = \tilde{G}(x,\omega)\exp{-i\omega x/c}, \tag{7}$$

the reduced propagation equation can be written in the form

$$\frac{\partial \tilde{E}(x,\omega)}{\partial x} = -i\frac{2\pi\omega}{c}\tilde{P}(x,\omega). \tag{8}$$

The spectrum of the polarisation response in the reduced wave Equation (8) is $\tilde{P}(x,\omega) = -N^{\text{at}}d_z(x,\omega)$, where the dipole moment $d_z$ in a spectral representation is calculated via the electron acceleration or the total force $f_z(x,\omega)$ acting on the electron as $d_z(x,\omega) = -f_z(x,\omega)/\omega^2$. Therefore, the polarisation response can be rewritten as

$$\tilde{P}(x,\omega) = N^{\text{at}}f_z(x,\omega)/\omega^2. \tag{9}$$

The choice to consider the polarisation through the total force $f_z$ gives the opportunity to naturally include the correct contribution of the free electrons due to the atomic potential and also to rigorously take into account the depletion of the electronic wavepacket .

To calculate the polarisation of a gas target, we calculate the expectation value of the time-dependent total force $f_z(x,t)$ as

$$f_z(x,t) = \tilde{E}(x,t) - \nabla V(\mathbf{r})\Psi(\mathbf{r},t), \tag{10}$$

where $\tilde{E}(x,t)$ is the inverse Fourier transform of $\tilde{E}(\omega,t)$ and $\Psi(\mathbf{r},t)$ is the wavefunction found via the numerical TDSE solution at the progressing propagation step.

We numerically solve the TDSE

$$i\frac{\partial}{\partial t}\Psi(\mathbf{r},t) = \left(-\frac{1}{2}\nabla^2 + V(\mathbf{r}) - \tilde{E}(x,t)z\right)\Psi(\mathbf{r},t)$$

for the model atom in the single-active electron approximation in the external field $\tilde{E}(x,t)$ (note that, here, $x$ is the field's propagation coordinate and $\mathbf{r}$ is the electron's coordinate; $z$ is the projection of $\mathbf{r}$). The equation is solved in cylindrical coordinates in the length gauge. A modified Coulomb potential [38] providing the ionisation energy equal to the one of argon is used:

$$V(r) = -\frac{1 + A\exp(-r)}{\sqrt{a^2 + r^2}},$$

with $A = 5.4$ and $a = 2.125$. See [30,38] and references therein for more details on the numerical method. The size of the numerical box for TDSE integration is chosen so that only the short electronic trajectories contribute to the microscopic response, as was performed in [39]; when suppressing the long trajectory contribution, we take into account the fact that, experimentally, this contribution produces a very divergent XUV beam, and thus does not contribute much to the observed XUV signal on the beam axis.

After finding $f_z(x,t)$, we find its Fourier transform $f_z(x,\omega)$ and then obtain the nonlinear polarisation response via Equation (9). We then substitute the polarisation response into the reduced wave Equation (8), which allows us to find the total field $\tilde{E}(x + \Delta x,\omega)$ after passing through a thin layer of matter with thickness $\Delta x$. This field is used again to obtain the nonlinear polarisation response by repeating this procedure in an iterative way for every new thin layer of medium.

We use a Ti:Sapphire laser pulse with wavelength 800 nm. Initially, the pulse has a $\sin^2$ envelope, and the initial carrier-envelope phase (CEP) corresponds to the sin-like pulse:

$$\tilde{E}(x = 0, t) = E_0 \sin^2\left[\frac{\pi t}{\tau}\right] \sin[\omega_0(t - \tau/2)],$$

$0 < t < \tau$. The pulse with such CEP has no zero-frequency component, as it should be for a real laser pulse. Both the pulse envelope and the CEP change during the propagation (but the zero-field component is certainly absent).

All simulations are performed for an argon target with density $3 \times 10^{18}$ cm$^{-3}$. Note that the 1D propagated results depend on the product of the length and the density, so the results can be attributed to other densities $N^{at}$ via multiplying the propagation distance presented in the results below by $3 \times 10^{18}/N^{at}$[cm$^{-3}$].

## 3. Results

First, we look at the temporal behaviour of the emitted XUV. Figure 1 shows the intensity of the XUV with a frequency above H30 generated by a fundamental with 10 fs duration (FWHM of intensity) as a function of time for three different propagation distances. One can see that after approximately 1.5 mm of propagation, an IAP is generated due to phase-matching gating.

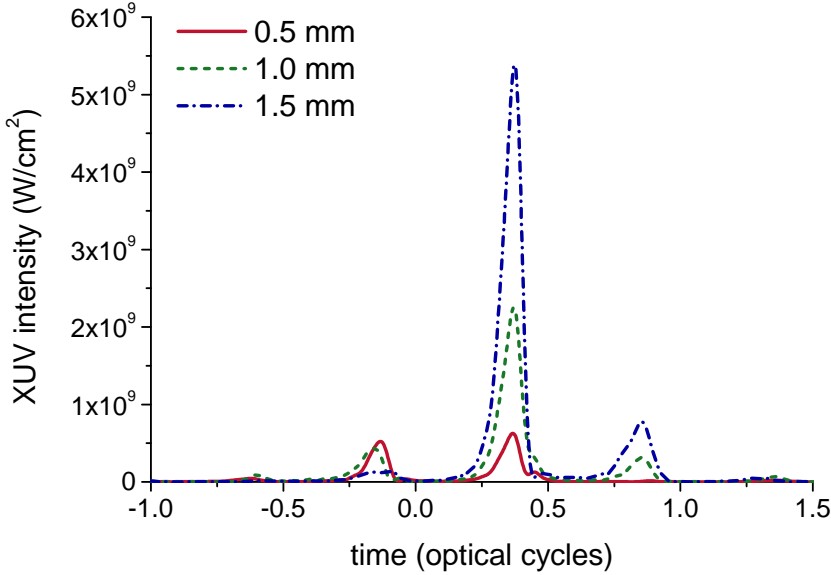

**Figure 1.** Attosecond pulses calculated for different propagation distances. The laser pulse duration is 10 fs, and its peak intensity is $2.6 \times 10^{14}$ W/cm$^2$.

To study the change in the temporal window of the phase-matching gating, we compare the temporal behaviour of the emitted XUV pulses, which are generated by laser pulses with different intensities and durations, during propagation. Note that previous studies [27] show that shorter pulses are preferable for IAP generation using phase-matching gating. The main reason is that, for a long pulse, the phase-matched generation is already achieved at the front of the pulse, i.e., under a relatively low laser intensity. For a short laser pulse, the temporal window of the phase-matched generation is close to the pulse maximum, i.e., occurs under a higher intensity. A higher intensity leads to rapid photoionization of the gas, and thus to a short temporal window. In the present study, we investigate the role of the pulse duration and intensity in more detail. Figure 2 shows the XUV intensity as a function of time and propagation distance for two different laser pulse durations and three different peak intensities. Each graph shows the XUV intensity with the photon energy above $I_p + 2U_p$, so this photon energy (and the harmonic order) is different for different

laser intensities: H22 for the laser intensity $1.4 \times 10^{14} \, \text{W/cm}^2$, H26 for the laser intensity $2.0 \times 10^{14} \, \text{W/cm}^2$ and H30 for $2.6 \times 10^{14} \, \text{W/cm}^2$. Note that Figure 1 presents three slices, shown in Figure 2c as vertical dashed lines.

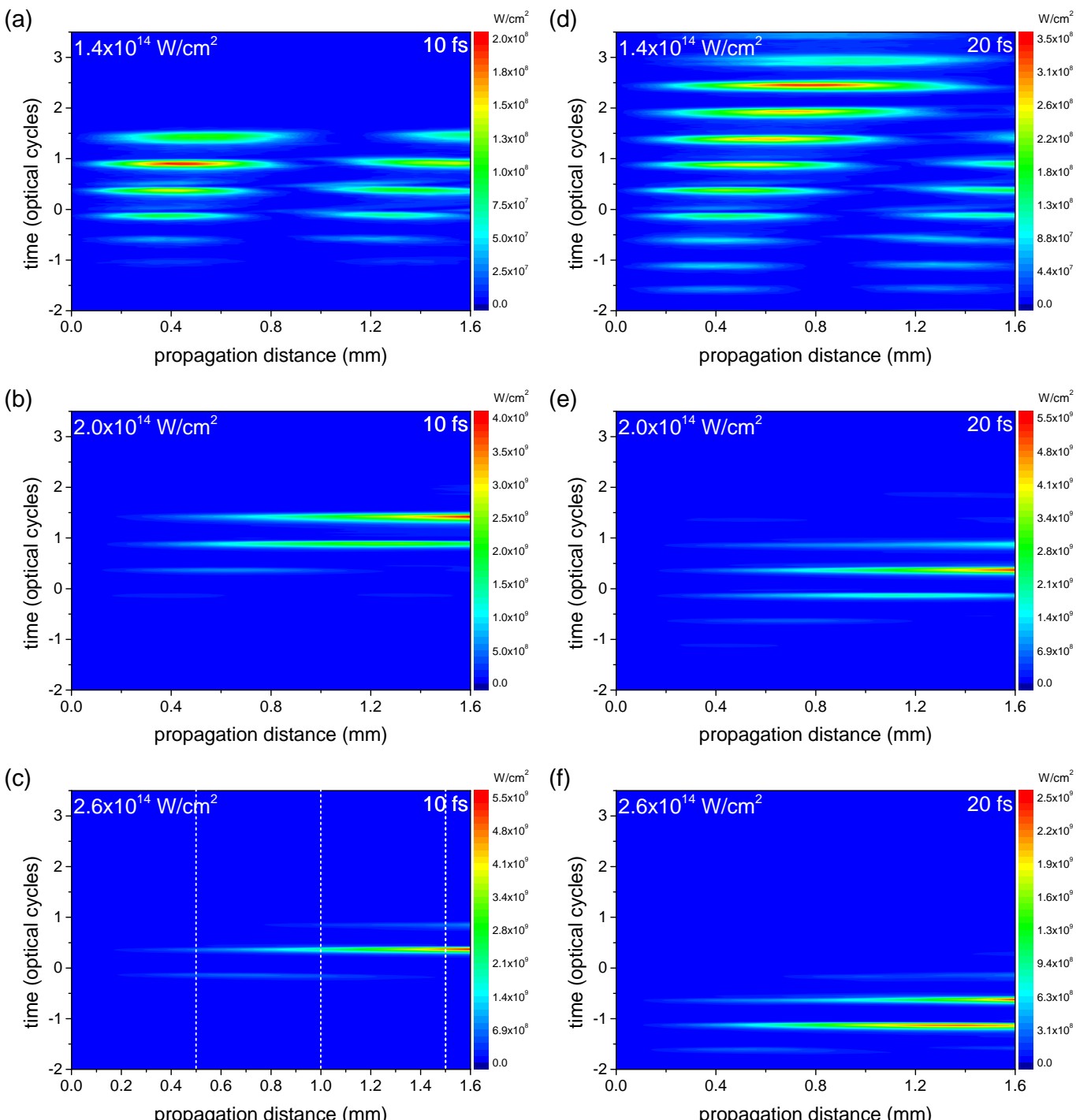

**Figure 2.** Intensity of XUV (with the photon energy above $I_p + 2U_p$) as a function of time and propagation distance. The laser pulse durations are 10 fs (**left** column, or (**a**–**c**)) and 20 fs (**right** column, or (**d**–**f**)). The laser pulse peak intensities are $1.4 \times 10^{14} \, \text{W/cm}^2$ (**upper** row, or (**a,d**)), $2.0 \times 10^{14} \, \text{W/cm}^2$ (**middle** row, or (**b,e**)) and $2.6 \times 10^{14} \, \text{W/cm}^2$ (**lower** row, or (**c,f**)). Note different scales of the colour bars.

One can see that with increasing laser intensity the number of emitted XUV pulses decreases, and the optimal conditions to produce IAPs in our case correspond to Figure 2c.

Now, let us consider the behaviour of the total XUV energy emitted as IAPs or near-IAPs for the most favourable intensity of the laser pulse and for 10 fs and 20 fs pulse durations. Figure 3 presents the XUV energy (the XUV intensity integrated over time) as a function of the propagation distance. The dependence of the high-harmonic energy on the propagation distance was studied in [30,31]. One can see in Figure 3 that, initially, the energy grows quadratically with the propagation distance. At a distance equal to the blue-shift length $L_{bs}$ (marked with arrows in the figure), the quadratic growth changes to a linear one.

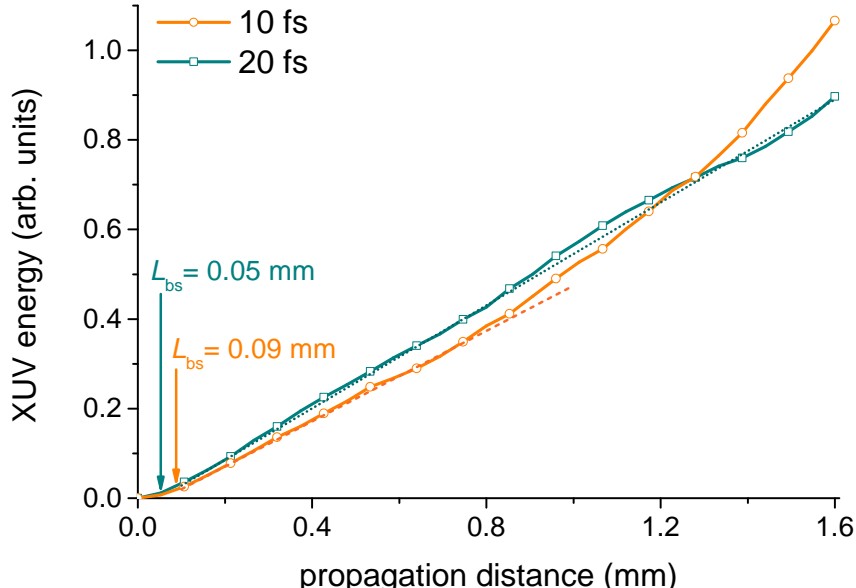

**Figure 3.** XUV energy as a function of propagation distance for the laser pulse durations 10 fs (orange curve with circles) and 20 fs (cyan curve with squares). The dashed lines present the linear trend. The laser intensity is $2.6 \times 10^{14}$ W/cm$^2$.

The linear growth can be explained in the temporal domain as follows. XUV is generated mainly within the phase-matching window, where the dispersion of the neutral gas is compensated by the free-electron one, appearing due to photoionisation. The XUV intensity within the phase-matching window grows quadratically with propagation distance, but the temporal width of the window decreases as the inverse of the propagation distance (assuming linear temporal variation in the fundamental refractive index near the centre of the window $t_0$, we have the HHG phase mismatch $\Delta\varphi(t) \propto (t - t_0)L$; if the window's temporal width is $\tau$, then $|\Delta\varphi(t_0 \pm \tau)| = \pi$; thus, $\tau \propto L^{-1}$), so the total XUV energy grows linearly.

In the spectral domain, this behaviour can be understood in the following way. The ionisation-induced blue shift of the fundamental leads to a blue shift of the harmonics in the spectrum of the microscopic polarisation induced by the total field (see Figure 4a). Thus, the spectral width of the propagated field of each harmonic, shown in Figure 4b, grows linearly with the propagation distance, and its spectral intensity is approximately constant. As a result, the total XUV energy growth is linear.

However, in Figure 3 one can see that for longer propagation distances, namely ones exceeding approximately 1.2 mm, the XUV energy for the 10 fs fundamental again grows faster than linearly, while for the 20 fs fundamental this is not the case. This behaviour can be explained in the temporal domain as follows. For this relatively long propagation distance, the phase-matching window becomes shorter than one half-cycle, so only one attosecond pulse, or IAP, is generated (see the dashed blue line in Figure 1). Further, the decrease in the window's temporal duration almost does not influence this attosecond pulse (until the durations of the attosecond pulse and the window become comparable).

The attosecond pulse intensity increases quadratically with the propagation length, and so does the XUV energy.

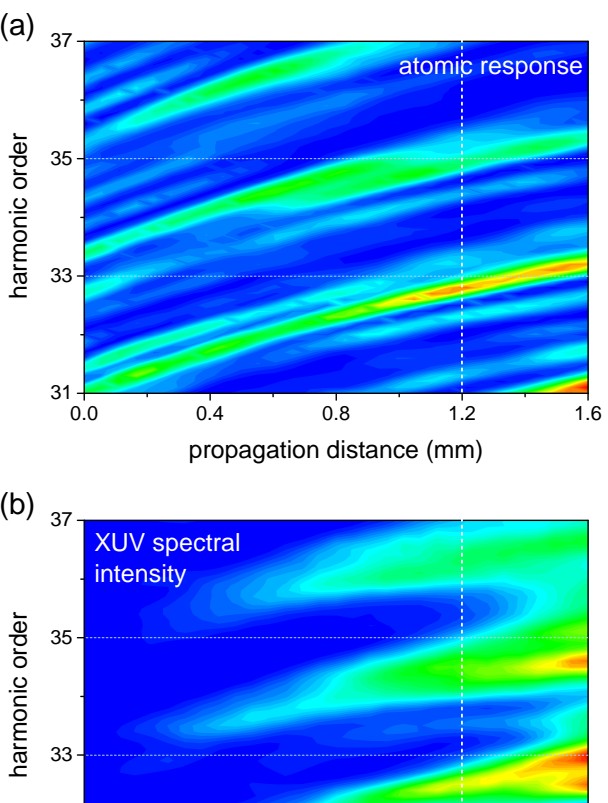

**Figure 4.** Spectra of the atomic response (**a**) and the propagated XUV field (**b**) as functions of propagation distance. The laser pulse duration is 10 fs, and its peak intensity is $2.6 \times 10^{14}\,\mathrm{W/cm^2}$. Colours present the signal from minimum (zero) in blue to maximum in red

Now, we look at the spectral domain vision of this behaviour. After approximately 1.2 mm of propagation (marked in Figure 4 with vertical dashed lines), the frequency shift of the atomic response is close to twice the laser frequency, so the response at harmonic $q$ generated at short propagation distances superimposes with the response at harmonic $(q-2)$ generated at long ones, as seen in Figure 4. The constructive interference of these signals leads to the quadratic growth of the XUV signal with propagation. We would like to stress again that this change from a linear growth to a quadratic one corresponds to the emission of a continuous spectrum, which results in the generation of an IAP. Note that in [30,31] we did not consider such long propagation distances leading to the overlap of the harmonic lines.

For even longer propagation distances, the XUV energy saturates and then decreases. In the temporal domain, this is due to further narrowing of the phase-matching window, which becomes comparable with the attosecond pulse duration. In the spectral domain, this situation corresponds to the accumulation of the phase mismatch of the contributions from different propagation lengths, and thus to their destructive interference. Note that our simulations underestimate the XUV absorption by the generating gas. In reality, the saturation would originate mostly from the absorption of the XUV.

Let us analyse the most favourable conditions for IAP generation. From Figure 2, we see that IAP generation requires a short fundamental pulse and high intensity (this

conclusion agrees with [27]). The high intensity is required to achieve a rapid variation in the refractive index due to the gas photoionisation. However, at first glace, it seems that the same variation of the refractive index can be obtained for a longer pulse (20 fs) as well, so it is not transparent why IAP generation is not achieved using the 20 fs fundamental.

In Figure 2, we can see that the phase-matching window shifts towards later times with propagation. To understand this feature in more detail, we present in Figure 5a the same results as in Figure 2c, and in Figure 5b we present the ionisation dynamics corresponding to the same conditions. One can see that the optimal ionisation degree (the one which compensates the neutral-gas dispersion) is achieved later for longer propagation distances —this explains the temporal shift of the phase-matching window. This behaviour of the ionisation is explained as follows: the laser pulse temporarily spreads while propagating due to the group velocity dispersion, so its peak intensity decreases. This leads to slower photoionisation at long propagation distances. Hence, small in comparison with the pulse duration, the temporal shift of the phase-matching window is comparable with the window duration. Note that this feature was not taken into account in [27], where the laser pulse envelope was assumed to be the same for different propagation distances. Now, we can conclude that the feasibility of IAP generation via phase-matching gating depends on the trade-off between the shortening of the phase-matching window and its temporal shift towards later times.

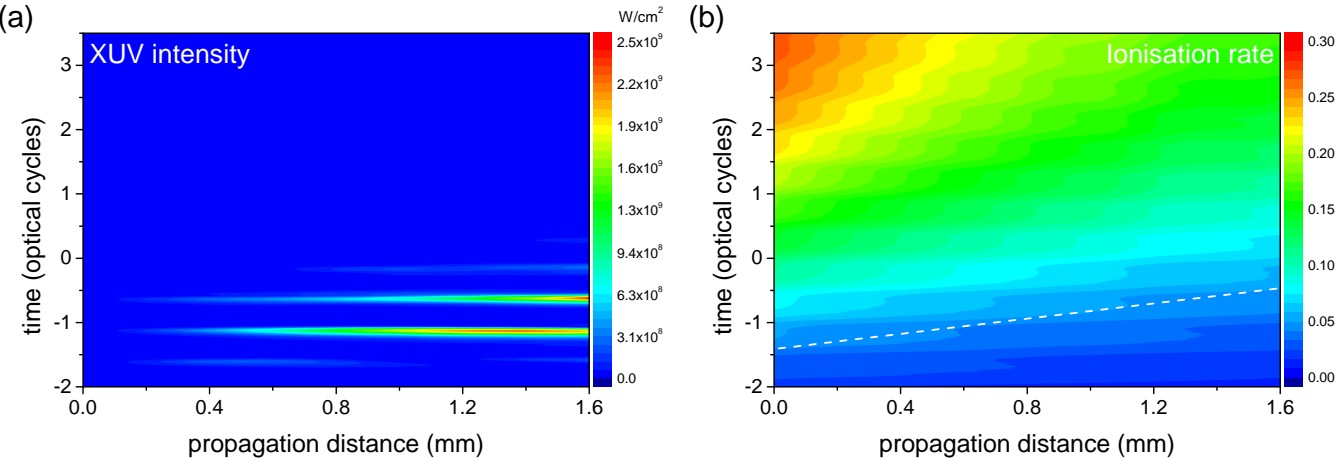

**Figure 5.** XUV intensity (**a**) and ionisation degree (**b**) as functions of time and propagation distance. The dotted line shows the position of the phase-matching window for XUV generation. The laser pulse duration is 20 fs and its peak intensity is $2.6 \times 10^{14}$ W/cm$^2$.

For shorter laser-pulse durations, the laser intensity within the window is higher, so the photoionisation takes place faster near the window, and thus the shortening of the window occurs faster than that for longer pulses. At first glance, it seems that the pulse spreading should also be more pronounced for the shorter pulse. This would be the case if the pulse spreading was given by the dispersion of the group velocity of a gas. However, this is not the case for our conditions. In our conditions, the front of the pulse propagates in the neutral gas with its group velocity. The falling edge of the pulse propagates in the ionised gas with another group velocity.

For the longer pulses, the ionisation degree is higher, so the relative spreading is approximately the same for the short and the long pulses. Thus, for the shorter pulses, the shortening of the window is faster and the temporal shift of the window is about the same as that for the long pulse. As a result, the short laser pulse provides better conditions for IAP generation. In Figure 2, we see that the 10 fs pulse provides such conditions, but the 20 fs laser pulse does not. The absence of IAP generation for the 20 fs laser pulse leads to the linear increase in the XUV intensity in Figure 3.

## 4. Discussion

In this paper, we consider the propagation effects which unavoidably accompany the generation of IAPs, even when the main gating technique used is different from the phase-matching one. For instance, in experiments where amplitude gating was achieved [5], 5 fs laser pulses were used. For such short pulses, the difference in phase matching for the generation at successive half-cycles can be pronounced, even for short propagation distances taking place in a gas jet. Moreover, many experiments (see, for instance, [40,41]) use much higher gas densities than the one we consider in this paper. This emphasises again that the propagation effects can play a crucial role even at short propagation distances.

## 5. Conclusions

In this paper, we study the production of an IAP due to transient phase-matching of HHG. For that, we numerically integrate the 1D propagation equation for the fundamental and generated fields with nonlinear polarisation calculated via the 3D TDSE solution. The use of the 1D propagation equation is a reasonable approximation to describe HHG with a flat-top laser beam.

We simulate the attosecond pulse generation with 800 nm fundamental pulses with 10 fs and 20 fs durations and with different intensities in argon gas. We find that IAP generation via phase-matching gating is achievable using the intense 10 fs laser pulse, and it is less feasible for the case of a 20 fs laser pulse for the considered intensities. The reason for this discrepancy is the temporal spreading of the fundamental pulse during the propagation because of the medium photoionisation. This nonlinear effect is defined by the difference in the group velocities for the neutral and photoionised medium. The spreading leads to the temporal shift of the phase-matching window towards later times for longer propagation distances.

Moreover, we study the behaviour of the XUV energy as a function of the propagation distance, and we find that for the 20 fs pulse this dependence is linear, but for the 10 fs one the onset of the IAP production corresponds to the change from linear to quadratic dependence of the XUV energy on the propagation distance. This feature can be used experimentally as a simple proxy method for the detection of IAP generation. Finally, it is important to note that the found dependencies on the propagation distance can be directly interpreted as dependencies on the medium pressure, as long as the 1D propagation picture is valid.

**Author Contributions:** Conceptualization, V.V.S.; methodology, V.V.S.; software, V.V.S.; validation, V.V.S. and M.A.K.; formal analysis, V.V.S.; investigation, V.V.S. and M.A.K.; resources, V.V.S.; data curation, V.V.S.; writing—original draft preparation, V.V.S.; writing—review and editing, V.V.S. and M.A.K.; visualization, V.V.S. and M.A.K.; funding acquisition, V.V.S. All authors have read and agreed to the published version of the manuscript.

**Funding:** The study was funded by RSF (grant No. 22-22-00242).

**Institutional Review Board Statement:** Not applicable

**Informed Consent Statement:** Not applicable

**Data Availability Statement:** Data will be made available on request.

**Acknowledgments:** We are grateful to Valeria Birulia for the optimisation of the propagation code. M.A.K. acknowledges support by the Royal Society through the University Research Fellowship URF\R1\231460

**Conflicts of Interest:** The authors declare no conflict of interest.

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
