# Peer review of "Phase-Matching Gating for Isolated Attosecond Pulse Generation"

_photonics, doi:10.3390/photonics10101122_

Round 1

Reviewer 1 Report

Manuscript entitled “Phase-Matching Gating for Isolated Attosecond Pulse Generation ” by V.V. Strelkov and M. A. Khokhlova presents a theoretical investigation based on numerical simulations related to the generation of isolated pulse generation (IAP) by controlling the phase matching phenomena in high-order harmonic generation (HHG) process. For this goal the authors propose the phase-matching gating technique, that is based on the creation of a temporal window for the macroscopically propagating HHG emission, due to the compensation of the phase mismatch of contributions coming from the free electrons and from the neutral atoms. This finite temporal window can be achieved by the relative contribution changes with the ionisation degree during the laser pulse. The authors claims that this window can be sufficiently short to fit a single attosecond pulse. 

For this proposal the authors perform numerical simulations by the numerical integration of the 3-dimensional time dependent Schrödinger equation (3D TDSE) on argon atoms including the 1D propagation effects. They perform the HHG spectra for two different laser pulse durations (10fs and 20 fs) as well as three different laser peak intensities. Authors analyze the harmonic intensity for some specific harmonic orders, as a function of the propagation distance along the pulse duration. They found that the degree of achievability for synthetize the IAP via phase-matching gating depends on the pulse duration due to the photoionization of the gas. Also is reported that the dependence of the XUV intensity with the propagation distance is linear for the longer pulse and quadratic for the shorter one. Author propose to use this fact as an indirect method in the IAP metrology.

The numerical methods used are correct and the result are consistent. The approach used (3D TDSE + 1D propagation) provides a complete description of the macroscopic harmonic field, which is a key parameter in the experimental measurements of the HHG spectra and attosecond pulses. 

The results reported on the manuscript can be useful for the HHG and Attosecond communities. I consider that the manuscript is suitable for publication in Photonics journal.

However below I stress some points that the authors should consider before publication:

1)    In spite that the authors give references about the approach used, some information is missing and should be briefly mentioned in the text. Regarding the 3D TDSE, what gauge is used, velocity gauge or length gauge? What coordinates are used, cylindrical, cartesian or spherical? Authors mention that a sin^2 laser pulse is used. However, the analytical form of the laser pulse would report more detailed information. In this sense, the 10 fs laser pulse represents, at 800 nm, a few cycles laser pulse. What about of the potential effects of the CEP (carrier envelope phase) in the attosecond temporal window for the attosecond pulse synthesis alluded?

2)    All the simulations are carried out using only one value of gas density (3x10^18 cm^-3).  A brief comment should be included why this value and why only one value of gas density has been used.

3)    Concerning the Figure 2, it represents the HHG intensity for selected harmonics orders, H22, H26, H30. In spite that these harmonics order are alluded in the text, but they are not mentioned in the caption of the figure. I suggest to include this information too in the caption of the figure, to avoid confusions, indicating what plot represents the corresponding harmonic order alluded in the text. Idem in Fig. 5.

4)    In the y-axis of Fig 3, what is plotted, XUV energy? or may be should be XUV intensity?

Author Response

We are grateful to the Referee for his/her positive report and for the useful comments.

_1)    In spite that the authors give references about the approach used, some information is missing and should be briefly mentioned in the text. Regarding the 3D TDSE, what gauge is used, velocity gauge or length gauge? What coordinates are used, cylindrical, cartesian or spherical? Authors mention that a sin^2 laser pulse is used. However, the analytical form of the laser pulse would report more detailed information. In this sense, the 10 fs laser pulse represents, at 800 nm, a few cycles laser pulse. What about of the potential effects of the CEP (carrier envelope phase) in the attosecond temporal window for the attosecond pulse synthesis alluded?

In the revised manuscript we essentially enlarge and amplify the Methods section. In particular, we present the items mentioned by the referee in this section.

_2)    All the simulations are carried out using only one value of gas density (3x10^18 cm^-3).  A brief comment should be included why this value and why only one value of gas density has been used.

We have added this comment in the end of the Methods section. 

_3)    Concerning the Figure 2, it represents the HHG intensity for selected harmonics orders, H22, H26, H30. In spite that these harmonics order are alluded in the text, but they are not mentioned in the caption of the figure. I suggest to include this information too in the caption of the figure, to avoid confusions, indicating what plot represents the corresponding harmonic order alluded in the text. Idem in Fig. 5.

We have modified the caption

_4)    In the y-axis of Fig 3, what is plotted, XUV energy? or may be should be XUV intensity?

 This is the XUV energy.

Reviewer 2 Report

In this manuscript, the authors investigated the production of an IAP using intense laser pulses to phase-match high-harmonic generation. They integrated propagation equations for fundamental and generated fields, considering nonlinear polarization from the time-dependent Schrödinger equation. They found that IAP generation is linked to a shift from a linear to a quadratic XUV energy-propagation distance relationship. They also showed that the maximum allowable duration of the fundamental pulse for IAP generation is limited by temporal spreading due to differences in group velocities between neutral and photoionized mediums. The result is interesting and worth to be published in Photonics, however, the following issues have to be addressed before it can be accepted.

 1. How does the intensity and duration of the fundamental laser pulses (e.g., 10 fs vs. 20 fs) affect the generation of IAP via phase-matching gating in argon gas, and what are the underlying physical mechanisms?

 2. How can the dependencies on propagation distance in this study be directly related to medium pressure, and what implications does this have for controlling and optimizing the generation of isolated attosecond pulses in different experimental setups?

 3. How does the temporal spreading of the fundamental pulse due to medium photoionization impact the generation of IAP, and how is this effect quantified?

 4. How does the medium photoionization affect the temporal spreading of the fundamental pulse during propagation and, consequently, IAP generation?

 5. A not-so-crucial suggestion, although the methods used in this paper have been described in previous articles, is still to recommend the author to briefly introduce them in this paper.

Author Response

We are grateful to the Referee for his/her positive report and for the useful comments.

_ 1. How does the intensity and duration of the fundamental laser pulses (e.g., 10 fs vs. 20 fs) affect the generation of IAP via phase-matching gating in argon gas, and what are the underlying physical mechanisms?

In the revised manuscript we enlarge and amplify the Methods section. In particular, in the beginning of this section we explain the physical mechanism of the transient phase-matching leading to the IAP generation. Moreover, in the Results section (page 4, left column) we explain the role of the pulse duration and intensity in the IAP generation via phase-matching gating

 _2. How can the dependencies on propagation distance in this study be directly related to medium pressure, and what implications does this have for controlling and optimizing the generation of isolated attosecond pulses in different experimental setups?

In the end of the Method section we comment on it

_ 3. How does the temporal spreading of the fundamental pulse due to medium photoionization impact the generation of IAP, and how is this effect quantified?

The temporal spreading leads to decrease of the peak laser intensity, thus to shift of the phase-matching window towards lower times. Hence small in comparison with the pulse duration, this shift can be comparable with the window duration. The shift of the window with the propagation leads to generation of several successive attosecond pulses. We have added the comment in page 7

_ 4. How does the medium photoionization affect the temporal spreading of the fundamental pulse during propagation and, consequently, IAP generation?

The refractive index appearing due to the free electrons affects the group velocity dispersion and thus leads to the spreading of the pulse. We have added the comment in page 7

_ 5. A not-so-crucial suggestion, although the methods used in this paper have been described in previous articles, is still to recommend the author to briefly introduce them in this paper.

Yes, in the revised manuscript we present much more detailed Methods section.

Round 2

Reviewer 1 Report

The manuscript “Phase-Matching Gating for Isolated Attosecond Pulse Generation ” by V.V. Strelkov and M. A. Khokhlova has been considerably improved on this second round.

The authors have properly addressed the points requested in my previous report.

I recommend the manuscript for publication on Photonics journal.